# Evaluation of Immune Status of Pigs against Classical Swine Fever for Three Years after the Initiation of Vaccination in Gifu Prefecture, Japan

**DOI:** 10.3390/pathogens13080616

**Published:** 2024-07-25

**Authors:** Keisuke Kuwata, Naotoshi Kuninaga, Yoko Kimura, Kohei Makita, Norikazu Isoda, Yukio Shimizu, Yoshihiro Sakoda

**Affiliations:** 1Gifu Central Livestock Hygiene Service Center, Gifu 501-1112, Japan; 2Veterinary Epidemiology Unit, Department of Veterinary Medicine, School of Veterinary Medicine, Rakuno Gakuen University, 582 Bunkyodai Midori-machi, Ebetsu 069-8501, Japan; kmakita@rakuno.ac.jp; 3Laboratory of Microbiology, Department of Disease Control, Faculty of Veterinary Medicine, Hokkaido University, Kita 18, Nishi 9, Kita-ku, Sapporo 060-0818, Japan; nisoda@vetmed.hokudai.ac.jp (N.I.);; 4One Health Research Center, Hokkaido University, Sapporo 060-0818, Japan; 5International Collaboration Unit, International Institute for Zoonosis Control, Hokkaido University, Kita 20, Nishi 10, Kita-ku, Sapporo 001-0020, Japan; 6Hokkaido University Institute for Vaccine Research and Development (HU-IVReD), Hokkaido University, Sapporo 001-0021, Japan

**Keywords:** classical swine fever, immune status, maternal antibody, sow, vaccination

## Abstract

In 2018, classical swine fever (CSF) reemerged in Gifu Prefecture, Japan, after 26 years of absence, and vaccination of domestic pigs using a live attenuated vaccine was initiated in 2019. Because the vaccine efficacy in piglets is influenced by the maternal antibody levels, vaccination should be administered at the optimal age by assuming the antibody level in sows. In this study, the shift in the antibody titer distribution in sows due to the initiation of vaccination to naïve herds and its influence on the vaccine-induced immunity rate in fattening pigs were investigated for 3 years. The results indicated that higher antibody titers were induced in first-generation sows after vaccine initiation because they were immunologically naïve, but the distribution of antibody titers shifted to lower levels along with their replacement with second-generation sows. The average vaccination age of fattening pigs became earlier year by year, and the vaccine-induced antibody rate was almost ≥80%. Based on the estimation of the optimal age for vaccination, it was found that vaccination at a younger age may reduce the risk of CSF infection. Taken together, the risk of CSF outbreaks can be reduced by administering vaccines at the optimal age based on the sequential monitoring of the sow’s immune status.

## 1. Introduction

Classical swine fever (CSF) is a disease among pigs and wild boars that is caused by the CSF virus (CSFV), which is classified as *Pestivirus suis* of the family *Flaviviridae* [1]. It is recognized as a particularly important infectious livestock disease because of its transmission intensity at a high density and the magnitude of the economic losses caused. In Japan, a live attenuated vaccine (GPE^−^ vaccine) was developed from a highly virulent strain classified as genotype 1 in 1969 [2] to control a continuous epidemic [3]. After the practical application of the vaccine, the outbreaks sharply decreased, and no outbreaks have been recognized since 1992. A national CSF eradication project was initiated, and all vaccinations were banned in 2006. Finally, Japan was recognized as a CSF-free country in 2015 by the World Organisation for Animal Health [4]. However, CSF reemerged at a pig farm in Gifu Prefecture in September 2018 [5]. After its reemergence in 2018, CSFV infection cases in wild boars were confirmed and transmission in wild boars caused CSF outbreaks on farms [5,6]. The national and prefectural governments and pig farms sought to prevent outbreaks by strengthening their biosecurity measures; however, the quick containment of the outbreaks failed. In this context, vaccination using the GPE^–^ vaccine for domestic pigs started in October 2019, approximately 1 year after the first outbreak in Gifu Prefecture, and had been expanded to 46 prefectures, excluding Hokkaido, as of September 2023 [7]. Although there have been no outbreaks on pig farms in Gifu Prefecture since the initiation of vaccination, outbreaks have continued through the spread of infection in wild boars in other areas, resulting in outbreaks in 33 cases in 14 prefectures in the vaccinated area [8,9]. One of the likely reasons for CSF’s expansion on pig farms could be inappropriate vaccination timing, because the pigs were infected before or just after vaccination in most cases [10]. Therefore, the immune status of the pig herd may be insufficient, and knowledge of the optimal age for vaccination will improve the situation of the pig herd.

The GPE^–^ live attenuated CSF vaccine currently used in Japan provides clinical protection 3 days after vaccination before neutralizing antibodies are detected 10–14 days after vaccination; these neutralizing antibodies are usually sustained for ≥3 years after single-dose administration [3]. The safety of this vaccine is ensured by the absence of clinical signs in vaccinated pigs, cohabiting infection, and the virulence reversion of the vaccine strain [3]. The most important factor influencing the efficacy of the live attenuated vaccine is the maternal antibody level at the time of vaccination. Generally, a nearly equal level of maternal antibody in piglets is transferred from their sows through the colostrum and gradually decreases as they grow [11]. If piglets with high maternal antibody titers are vaccinated, the maternal antibodies interfere with the development of vaccine-induced antibodies [11]. Nonetheless, in nonvaccinated piglets with maternal antibodies, a neutralizing antibody titer of ≥1:32 should be required to protect against CSF [12]. Ideally, ≥80% of the herd immunity rate is sufficient to reduce the risk of CSF in the herd, including immunized and nonimmunized individuals. Thus, it is desirable to vaccinate piglets at the optimal age to maintain the protective status in herds by monitoring maternal and vaccine-induced antibodies. It has been proven through CSF eradication in Japan that CSF can be controlled if the herd immunity rate by sufficient maternal and vaccine-induced antibodies is ≥80% [3].

To achieve this criterion, piglets should be vaccinated at the optimal age, with the distribution of neutralizing antibody titers in sow herds considered an important indicator [13]. This is because piglets inherit the same level of maternal antibody titers via the colostrum, showing a linear decrease on a log scale with a half-life of approximately 11 days after birth [14]. In the instruction leaflet of the commercial GPE^–^ vaccine, it is described that vaccination should be administered once between 30 and 60 days of age, taking into consideration the maternal antibodies of the sow. Furthermore, based on the results of the antibody statuses of sows in the field in the past, the recommended vaccination period is between 30 and 40 days of age [3]. Sows are vaccinated a total of four times according to the vaccination program, with the same initial vaccination as fattening pigs, six months after this, and then every other year. Although there are no regulations on the vaccination program for pre-mating and pre-farrowing vaccination, the trend is to avoid these vaccinations. During the first round of vaccination in 2019, all pigs in Japan were immunologically naïve to CSFV, and first-generation sows (SOW-G1) were vaccinated. Immunologically naïve pigs in 2019 showed a stronger immune response to vaccination than in 1999, when domestic pigs had been consecutively vaccinated since 1969 [15]. Furthermore, second-generation sows (SOW-G2) delivered from SOW-G1 inherited high maternal antibody levels, and the immune responses after vaccination were different between SOW-G1 and SOW-G2, even after the implementation of the same vaccination program [16]. The gradual generation change from SOW-G1 to SOW-G2 on the farm may result in the high diversity of the immunity levels in sows and piglets delivered from these sows with various immune levels.

As establishing the optimal vaccination program is important for the effective prevention of CSF outbreaks in Japan, it is necessary to clarify the immunity status in pig herds after the restart of CSF vaccination. In this study, the distribution of neutralizing antibody titers by generation in sows was investigated after vaccination in Gifu Prefecture with immunization status in fattening pigs. Based on these data, the optimal age for the vaccination of piglets according to the different antibody statuses of sows was estimated, and the risk of CSF infection in the vaccinated fattening pig herd was assessed.

## 2. Materials and Methods

### 2.1. Serum Samples

Blood sampling on the farm was performed by veterinarians who were technically trained in accordance with the administrative direction of the Ministry of Agriculture, Forestry and Fisheries of Japan. Serum samples were collected from 1452 sows at 19 farms and 1982 fattening pigs at 24 farms in Gifu Prefecture from 2020 to 2022. Sow samples were collected from breeding pigs ≥ 1 year old, and fattening pig samples were collected from pigs ≥ 90 days post-vaccination and ≥120 days old with no remaining maternal antibodies. Information on the birth date, vaccination date, and serum sampling date of each pig was recorded.

### 2.2. Pig Herd Vaccination

Pig vaccination using the GPE^−^ vaccine on farms was performed by veterinarians who were technically trained, as with the blood sampling. Sows of all generations were vaccinated for the first time at approximately 30–60 days of age, for the second time six months later, and for the third and fourth time every year thereafter. All fattening pigs were vaccinated at approximately 30–60 days of age with only one vaccine dose.

### 2.3. Antibody Detection Methods

For the sow samples, a serum neutralization test (SNT) using CPK-NS cells was conducted according to the method described by Sakoda et al. [17]. For the fattening pig samples, an enzyme-linked immunosorbent assay (ELISA) was conducted as a screening test using a CSF ELISA kit II (Nippon Gene Corp., Tokyo, Japan) [18], according to the manufacturer’s instructions. This ELISA is an indirect ELISA and correlates with the SNT because the E2 glycoprotein is the main component, but the lower limit of detection for this ELISA is 1:16 [18]. Because of the lower sensitivity of the ELISA compared to the SNT, ELISA-negative samples (S/P value < 0.05) were retested using the SNT to confirm the low antibody titers. Because the SNT is a complex procedure, a first screening using a convenient ELISA and then performing the SNT on ELISA-negative samples is efficient. The SNT was performed with an initial serum dilution of 1/1 using serially diluted serum, and the SNT titers were expressed as the reciprocal of the dilution fold of the serum.

### 2.4. Classification of Sows and Sequential Evaluation of Antibody Levels

During the 3-year study, the period from April to September was referred to as the first term of the corresponding year, and the period from October to March of the next year was referred to thereafter as the second term. For each term, the distribution of the SNT titers of the sows was indicated. In Gifu Prefecture, the vaccination of all pigs started simultaneously on 25 October 2019. Thus, SOW-G1 and SOW-G2 were categorized as sows born on or before 25 October 2019 and on or after 25 January 2020, respectively. Sows born between 25 October 2019 and 25 January 2020 were categorized as generation 1.5 (SOW-G1.5). The distributions of the SNT titers for SOW-G1, SOW-G1.5, and SOW-G2 were indicated by avoiding double counting, because the samples may have been collected from the same individuals throughout each term. The Wilcoxon rank-sum and Kolmogorov–Smirnov tests were performed to compare the mean and distribution of the SNT titers in SOW-G1, SOW-G1.5, and SOW-G2.

### 2.5. Sequential Evaluation of Antibody Levels in Fattening Pigs

To confirm the effectiveness of the CSF vaccine over the three-year period, the ELISA-positive rate of fattening pigs per term was calculated for each farm. The vaccine-induced antibody rate was defined as the rate of the total number of either ELISA-positive or ELISA-negative but SNT-positive samples per number of fattening pigs tested on the farm. These positive rates for each term were averaged for all farms by each term. Eighty percent was defined as a sufficient vaccine-induced antibody rate based on a previous report [3], and the number of farms achieving it was counted. The antibody titer of each sample was determined using the SNT for ELISA-negative samples. In addition, the SNT titers were estimated for ELISA-positive samples based on the S/P value of the ELISA according to a previous report [19]. These individual data were averaged for all farms for each term. The age of the pigs at vaccination was averaged per term. The antibody-positive rate was independently confirmed for the farms vaccinated at 30 days of age or earlier, because vaccination at ≤30 days of age was recommended based on the immune status of sows, regardless of the general optimal age for vaccination (30 to 40 days of age) [3].

### 2.6. Estimation of Optimal Age for Vaccination

The optimal age of piglets for vaccination was estimated based on the distribution of the SNT titers of sows throughout Gifu Prefecture. The simulation was constructed using the calculation concept proposed by Shimizu [3]. In brief, the immunity of piglets after birth was categorized into three phases: maternal antibody transfer from sows to litters, maternal antibody decay in piglets, and the immune response in piglets after vaccination. The simulations were as follows. The antibody titers of sow herds were measured by the SNT, and a distribution was created. From this distribution, the maternal antibody titer per age of the piglet herd was calculated, assuming that the antibody titers at sow and piglet birth were the same, the half-life of the maternal antibody after birth was 11 days [14], and the protective maternal antibody titer was ≥1:32 [12]. Under these parameters, the percentage of piglets with protective maternal antibodies remaining every 11 days after birth was calculated. The vaccine-induced antibody was simulated to be induced at 100% for piglets with maternal antibody titers of ≤1:64, 50% for those with 1:128 to 1:1024, and 0% for those with ≥1:2048 at vaccination [3]. Herd immunity of ≥80% is essential to protect herds from CSF. In this model, the optimal age of the piglet for vaccination was determined as the minimum age providing ≥ 80% of the vaccine-induced antibody rate. The percentage of piglets possessing sufficient maternal antibody titers to protect them from CSF at this age was estimated.

Using this method, the optimal age for vaccination on 45 fattening pig farms was estimated individually based on the SNT data of sows six months prior.

### 2.7. Evaluation of Risk of CSF Infection in Fattening Pig Herds

The risk of infection in fattening pigs was estimated based on the distribution of the SNT titers of sows and the estimated optimal age for the vaccination of piglets. Vaccine-induced immunity, including antibody- and cell-mediated immunity, protects pigs from CSF 5 days after vaccination [3,20]. In this model, fattening pigs with retained maternal antibodies were defined as protected at ≥1:32 antibody levels, and fattening pigs with vaccine-induced immunity were defined as protected from 5 days after vaccination. The proportion of days < 80% of the herd immunity rate throughout the lifetime of the fattening pigs (defined as 180 days in this study) was calculated as the risk value using the following equation:$$Risk=\frac{\sum_{t=0}^{d}\left(0.8-p_{t}\right)I\left(p_{t}<0.8\right)}{d}$$
where $p_{t}$ is the herd immunity rate at day age *t*, *d* is the number of days from birth to slaughter, and I($p_{t}$ < 0.8) is a state function of 1 when *p_t_* < 0.8 and 0 otherwise. In each of the six terms (6 months), the risk values when the CSF vaccine was administrated at 35 days and at the optimal piglet age at vaccination were compared. Thirty-five days for vaccine administration was selected as a representative value, given that the optimal age at vaccination for piglets is generally set at 30 to 40 days of age [3]. These analyses were conducted using Microsoft Excel (Microsoft 365 MSO, version 2302 Build 16.0.16130.20690).

## 3. Results

### 3.1. Investigation of Antibody Distribution in Sows

To determine the status of maternal antibody retention in piglets at the first vaccination, the sows used in the study were divided into three generations: SOW-G1, SOW-G1.5, and SOW-G2. They were monitored every 6 months for 3 years. At the start of vaccination, 100% of sows were classified as SOW-G1, but this percentage declined year by year. Alternatively, the percentage of SOW-G2 in the second term of 2021 was 50.7% and reached 87.2% in the second term of 2022 at the end of the study (Table 1 and Appendix A).

The SNT titers of sows after vaccination have been measured since 2020, and the distributions of the SNT titers every 6 months were compared (Figure 1a–c). Based on these results, the basic statistics of the distributions were calculated (Table 2 and Appendix A). The mean SNT titer decreased from 1:2^9.1^ in the first term of 2020 to 1:2^5.4^ in the second term of 2022, and the median and mode decreased from 1:2^9^ to 1:2^6^ throughout the examination period. The log_2_ of the standard deviation (SD) increased from 1.6 in the first term of 2020 to 2.3 in the second term of 2022 (Table 2).

To confirm the immune status of the sows by generation, the distribution of the SNT titers was compared among each sow generation (Figure 1d). The mean, median, and mode of SOW-G2 were lower, and the SD was higher than that of SOW-G1 (Table 2). The mean (or distribution) values of SOW-G1, SOW-G1.5, and SOW-G2 were significantly different according to the Wilcoxon rank-sum (or Kolmogorov–Smirnov) test (*p* < 0.05).

### 3.2. Investigation of Vaccine-Induced Antibodies in Fattening Pigs

Changes in the immune response due to vaccination in sows may affect fattening pigs. Based on the ELISA and SNT results using fattening pig serum, the antibody-positive rate by CSF vaccination at fattening farms (14–21 farms) was investigated every 6 months between 2020 and 2022. The average vaccination age of fattening pigs in each term was 42.4 days in 2020 and 47.1 days in the first term of 2021 and gradually became earlier thereafter, being 37.9 days in the second term of 2022. The ELISA-positive rate never reached 80% in all terms (Table 3, Appendix A). In contrast, the vaccine-induced antibody rates, including those tested positive by ELISA and SNT, were always >90% (Table 3, Appendix A). Three farms had vaccine-induced antibody rates of <80%: 78.6%, 78.0%, and 72.7% (Appendix A). Four farms were vaccinated before 30 days of age, and all had vaccine-induced antibody rates of >80%: 96.7%, 83.3%, 83.3%, and 86.7% (Appendix A). The mean antibody titers of fattening pigs at each term ranged from approximately 1:16 to 1:128 (Table 3, Appendix A).

### 3.3. Estimation of Optimal Age for Vaccination and Risk of CSF Infection in Fattening Pig Herds

The antibody titers in sows reflect the maternal antibody levels in piglets. This indicates that piglets should be vaccinated in consideration of the antibody titers of sows. Therefore, the optimal piglet age at vaccination under the different immunological statuses of the herd in Gifu Prefecture was simulated considering the SNT titer distribution in sows (Figure 1). It was estimated that the optimal ages at vaccination, expected to lead to a vaccine-induced antibody rate of ≥80% by vaccination in each term between 2020 and 2022, were 34, 37, 24, 18, 9, and 0 days, respectively (Figure 2). The rates of piglets carrying protective maternal antibodies at the optimal age at vaccination were 84%, 83%, 73%, 67%, 76%, and 67% in each term between 2020 and 2022. After the first term of 2021, the proportion of piglets carrying protective maternal antibodies was <80% for each investigation (Figure 2c–f). 

The optimal age for vaccination was estimated for 45 farms with sow data. It was evident that most farms were vaccinated later than the optimal age (Appendix A). Nevertheless, three farms were vaccinated earlier than the optimal age, and one farm was vaccinated with a gap of 1 day from the optimal age. The vaccine-induced antibody rates at these four farms were ≥80% (Appendix A).

From these maternal and vaccine-induced antibody data throughout Gifu Prefecture, the proportion exhibiting <80% of the ideal herd immunity rate was calculated as the risk value (Figure 3). The risk values for the vaccination of piglets at 35 days of age gradually increased from 0.0% in the first term of 2020 to 9.0% in the second term of 2022. In contrast, the risk values when applying the optimal age at vaccination in each term were sustained at the lowest level during the study period, between 0.0% and 0.8%. 

## 4. Discussion

CSF outbreaks in domestic pigs in Japan have decreased dramatically since vaccination initiation in 2019. However, Japan is the only country that has eradicated CSF once, stopped vaccination, and initiated vaccination again, and it was unclear what the immune status would be. It is necessary to clarify the difference in antibody responses during constant vaccination and immediately after vaccination initiation to develop the optimal vaccination program. Therefore, the current study was conducted to understand the efficacy of vaccination, including maternal and vaccine-induced antibodies in different pig generations, and to estimate the optimal age for vaccination and the risk based on the immunity level. For this purpose, a quantitative estimation method for the optimal age for vaccination was established and applied to evaluate the immune status of pigs after vaccination. Our results demonstrated that a gradual shift in the antibody level distribution in sow herds may affect the risk of CSF introduction into herds and the ideal age for piglets to be vaccinated. 

The proportion of SOW-G1 was 100% at the start of vaccination, but half were replaced with SOW-G2 after 2 years and >80% with SOW-G2 after 3 years (Table 1). A previous study suggested that it would take 2 to 3 years to replace the next-generation sows based on the current renewal cycle in pig farming operations in Japan [21], which is consistent with our findings. In this study, the median and mode of antibody titers in SOW-G1 were eightfold higher than those in SOW-G2 (Table 2). This is probably because SOW-G1 were vaccinated without maternal antibodies to CSFV, which allowed the vaccine virus to multiply efficiently in the pigs’ bodies, resulting in a strong immune response. The variation in the antibody distribution of sows gradually increased from 2020 to 2021, with the SOW-G2 numbers exceeding those of SOW-G1 (Table 2). In the SOW-G1 versus SOW-G2 comparisons, SOW-G1 had a higher median and lower SD (Figure 1d). This can be explained by the fact that strong immune responses were induced by vaccination in most SOW-G1, which were immunologically naïve and a uniform population, whereas the inducement of immune responses in SOW-G2 was divergent, making it difficult to estimate the optimal age at the vaccination of piglets.

The rate of the vaccine-induced antibody in fattening pigs to confirm vaccine efficacy did not reach 80%, which should be the sufficient herd immunity level, by ELISA alone, although the positive rate was >90% in the SNT (Table 3). The ELISA, which is currently used in Japan, is less sensitive than the SNT in detecting antibodies to CSFV, and the detection limit in the ELISA was regarded as 1:16 of the SNT titer [18]. When pigs with an SNT titer of ≥1:32 by CSF vaccine are challenged with virulent CSFV, they do not excrete viruses, but an immune response occurs, establishing so-called sterile immunity [22,23]. However, in pigs with SNT titers ≤ 1:16, transient signs such as fever were observed following virus challenge, although it was confirmed to be effective in reducing disease onset [22,23]. In ELISA-negative but SNT-positive cases, the pig can be recognized as immunized with the vaccine. Thus, it is necessary to use not only an ELISA but also the SNT to determine the rate of vaccine-induced antibodies on farms. The overall vaccine-induced antibody rates were excellent but did not exceed 80% at the three farms (Appendix A). Porcine reproductive and respiratory syndrome virus and circovirus type 2 cause immunosuppression and inhibit the efficacy of the CSF vaccine [24,25,26,27]. Considering the prevalence of these infectious diseases in Japan, it is necessary to elucidate the influence of these infectious diseases on vaccination and maternal antibodies against CSFV as factors influencing the efficacy of CSF vaccination. In contrast, the vaccine-induced antibody rate on four farms exceeded 80%, although the vaccine was administered at 20 to 30 days of age (Appendix A). It is possible to vaccinate earlier than the general vaccination age.

Estimating the optimal age of piglets for vaccination based on the distribution of the SNT titers in sows, the maternal antibody protection rate of piglets decreased each year (Figure 2). The optimal age of piglets for vaccination was 30 to 40 days in 2020, but it shifted to 10 to 30 days in 2021 and <10 days in 2022 (Figure 2). The cause of this change was decreased antibody titers due to the generational change in sows. The risk values of vaccination at 35 days of age increased year by year (Figure 3). In contrast, this risk could be improved by vaccination at an early and optimal age (Figure 3). However, the maternal antibody protection rates were <80%, even when vaccinated at the optimal age. This indicates that even if the vaccine was administered at the optimal age, the risk was inevitable due to the varying antibody distribution of the sows, consistent with CSF infection cases occurring at a prevaccination age of days [10]. Considering the high vaccine-induced immunity in SOW-G1, there has been a tendency to delay vaccination for piglets born from SOW-G1 beyond the optimal age for vaccination in Japan. However, the results of this study indicate the necessity to advance the timing of vaccination prior to the traditionally recommended 30–40 days. Furthermore, the variation in the immune status of SOW-G2 sows complicates the determination of the optimal age for vaccination. As described in a previous study [16], the temporary use of two vaccination doses during this period may represent a countermeasure to minimize the risk of infection.

The method of calculating the optimal age for vaccination based on a previous report [3], which contributed to the eradication of CSF in Japan, was applied to the current antibody situation. Between 2020 and 2022, many farms were vaccinated late, considering the first generation, and the vaccine-induced antibody rate was ≥80% and close to 100%. Most importantly, three farms were vaccinated at an age earlier than the estimated optimal age, and one farm was vaccinated at the same time as the optimal age, all achieving vaccine-induced antibody rates of ≥80% (Appendix A). Thus, we can highlight some evidence to support our estimation of the optimal age for CSF vaccination. However, the number of these practices is small and insufficient to support our assertion. Therefore, it is necessary to increase the number of practices in the future. Challenges with CSF vaccination prior to 30 days of age include overstress when vaccinating young pigs and coordination with vaccination programs for other viral diseases. These factors need to be resolved when accumulating data on the evaluation of the optimal vaccination timing. On the other hand, if it is possible to raise the antibody titers of sows as a herd, the optimal age for vaccination would be returned to 30–60 days of age. As a feasible countermeasure, the repeated vaccination of sows should be implemented, although the booster effects of the GPE^−^ vaccine (attenuated live vaccine) with repeated injections are limited. Alternatively, additional immunological stimulation with an inactivated vaccine may be feasible in the future, although no approved inactivated vaccines are currently available in Japan.

Through these investigations, the change in the antibody distribution of sows after the initiation of vaccination in Gifu Prefecture and the change in the optimal age of vaccination of piglets have become clear. Ideally, the optimal age for vaccination should be estimated on a sow-by-sow or farm-by-farm basis. As a first trial, this study was conducted to estimate the optimal age for vaccination throughout the Gifu Prefecture area, because vaccination in Japan in 2019 was restarted on a prefecture-by-prefecture basis. This approach is also expected to be applied to sow-based and farm-based vaccination programs at the optimal age for vaccination. Once these data are accumulated, the determination of the optimal age for vaccination may be available for SOW-G1 without immunity testing because of the uniform SNT titer without variation; however, for SOW-G2, the optimal age for vaccination should be confirmed by antibody testing because of the variation in SNT titers. 

Other concerns include the low protection by maternal antibodies against field CSFV with diverse antigenicity and moderate pathogenicity. In this study, the protection level of the maternal antibody was set at ≥1:32. However, because of the different genotypes of the GPE^−^ strain and epidemic strain and the pathogenicity of the causative field virus [5], this value may be fixed at a higher SNT titer [23]. Based on the above, it is necessary to accumulate more data on CSF vaccination and to implement more effective CSF countermeasures.

## 5. Conclusions

The seronegative sow population that had never been vaccinated, i.e., the first generation, showed a relatively stronger immune response. The second-generation sows derived from the first-generation sows had lower antibody titers after vaccination due to interference by the maternal antibody. The distribution of antibody titers in the sow population was different between the first and second generations. This study established a quantitative estimation method to determine the optimal age for vaccination based on herd immunity formation and a risk assessment of infection. Using this method, the immune status of the pig herd was clarified after CSF reemergence and revaccination. This method will be applied to evaluate the immune status of pigs after the initiation of vaccination.

## Figures and Tables

**Figure 1 pathogens-13-00616-f001:**
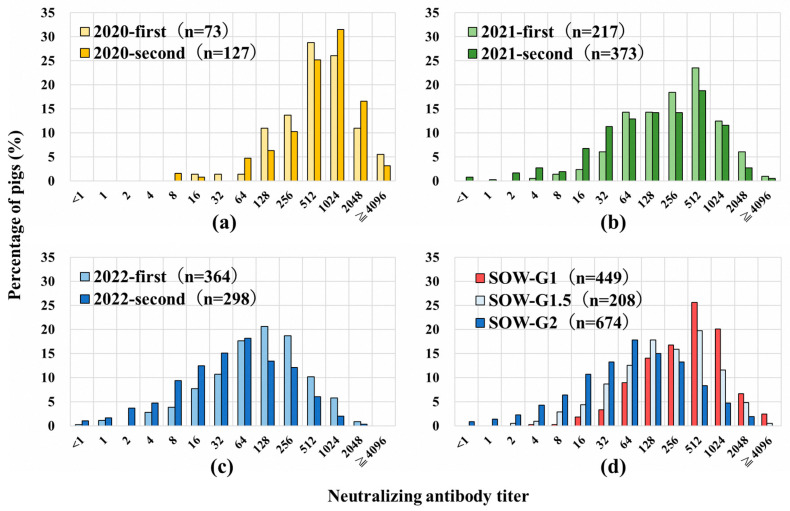
Distribution of antibodies against classical swine fever virus (CSFV) in sows. Serum neutralization test (SNT) titers in sows after vaccination were measured, and the distribution of SNT titers in sows in the first and second terms of (**a**) 2020, (**b**) 2021, and (**c**) 2022 is indicated. (**d**) Distribution of SNT titers according to the classification of sows in generation 1 (SOW-G1), 1.5 (SOW-G1.5), and 2 (SOW-G2).

**Figure 2 pathogens-13-00616-f002:**
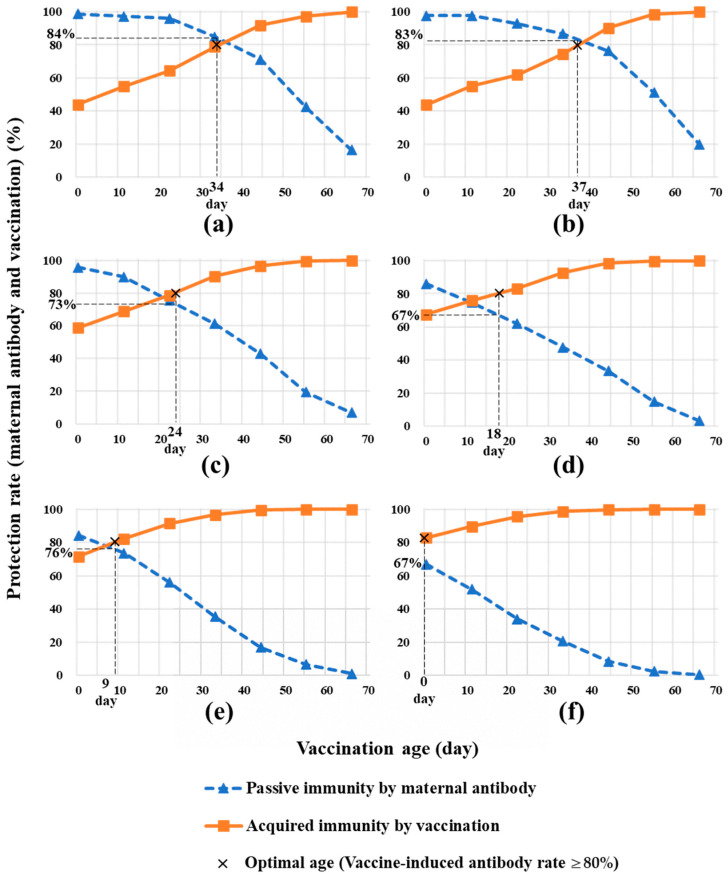
Estimation of the optimal age at classical swine fever (CSF) vaccination. Based on the serum neutralization test titer distribution of sows, the rate of piglets with maternal antibody protection and the rate of immune response by vaccination in piglets were simulated at 11-day intervals. The optimal age for vaccination is the youngest age at which the vaccine-induced antibody rate is ≥80%. The percentage of piglets with protective maternal antibodies at each estimated optimal age for vaccination is also indicated (X). (**a**) 2020-first, (**b**) 2020-second, (**c**) 2021-first, (**d**) 2021-second, (**e**) 2022-first, and (**f**) 2022-second terms.

**Figure 3 pathogens-13-00616-f003:**
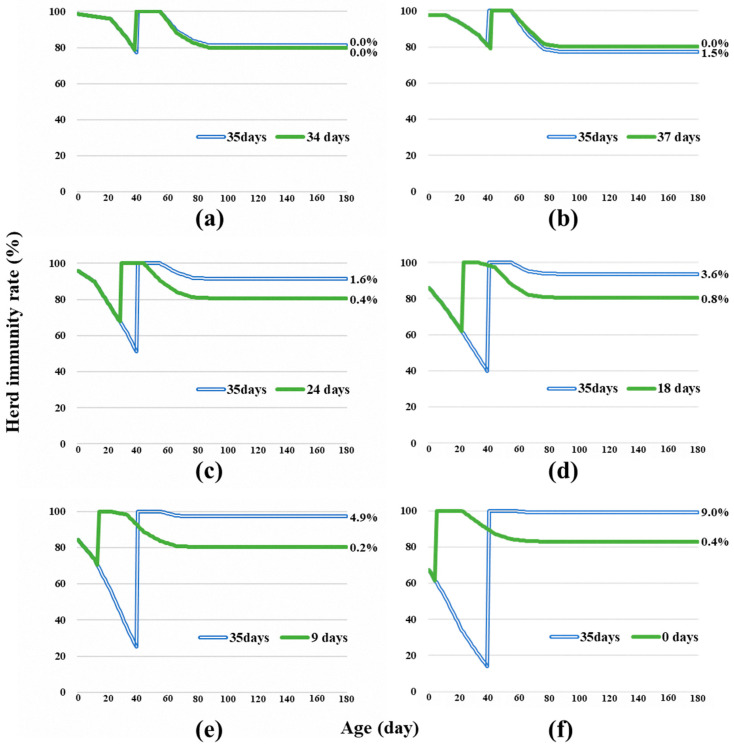
Herd immunity rates of the pig population vaccinated at an estimated optimal piglet age at vaccination for classical swine fever (CSF). The blue double line represents the herd immunity rate when vaccinated at the standard age (35 days of age), and the green line represents vaccination at the optimal age simulated in Figure 2 for each term. The values on the right of the line indicate the risk values, representing the proportion of days <80% of herd immunity throughout the lifetime (180 days). (**a**) 2020-first, (**b**) 2020-second, (**c**) 2021-first, (**d**) 2021-second, (**e**) 2022-first, and (**f**) 2022-second terms.

**Table 1 pathogens-13-00616-t001:** Number of sows for each generation every 6 months from 2020 to 2022.

Generation of Sows	No. of Sows in Each Period ^1^						Total No.
	2020-First	2020-Second	2021-First	2021-Second	2022-First	2022-Second	
SOW-G1	73	102	91	120	81	20	487
SOW-G1.5	0	22	75	64	48	18	227
SOW-G2	0	3	51	189	235	260	738
Ratio of SOW-G2 in Total ^2^ (%)	0	2.4	23.5	50.7	64.6	87.2	50.8

^1^ The antibody survey was performed six times for 3 years every 6 months. ^2^ SOW-G2/(SOW-G1 + SOW-G1.5 + SOW-G2). First: April to September; second: October to March next year.

**Table 2 pathogens-13-00616-t002:** Basic statistics of neutralizing antibody titers of sows inoculated with classical swine fever vaccine.

	Antibody Titer in Each Period ^1^ (log_2_)						Antibody Titer in Each Generation (log_2_)			
	2020		2021		2022					
							SOW-G1	SOW-G1.5	SOW-G2	
	First	Second	First	Second	First	Second				
Mean	9.1	9.2	7.9	7.1	6.6	5.4	8.4	7.5	6.0	
Median	9	10	8	7	7	6	9	8	6	
Mode	9	10	9	9	7	6	9	9	6	
SD	1.6	1.7	1.9	2.4	2.1	2.3	1.8	2.1	2.4	

^1^ The antibody survey was performed six times for 3 years. First: April to September; second: October to March next year. SD: standard deviation.

**Table 3 pathogens-13-00616-t003:** Antibody-positive rate in fattening pigs inoculated with the classical swine fever vaccine.

Term ^1^	Farm No.	Average Vaccination Age (Day)	Pig No.	ELISA- Positive Rate (%)	Vaccine-Induced Antibody Rate ^2^ (%)	Mean Antibody Titer ^3^ (log2)	Farm No. with ≥80% Antibody- Positive Rate
2020- second	14	42.4 [35–60] ^4^	401	71.2 [46.5–100] ^4^	90.2 [78.6–100] ^4^	5.5 [3.0–10.0] ^4^	13/14
2021- first	16	47.1 [31–60]	397	72.5 [33.3–100]	97.1 [84.6–100]	5.5 [2.6–10.5]	16/16
2021- second	14	42.4 [33–51]	292	58.8 [15.0–90.0]	92.7 [85.0–100]	4.1 [1.3–6.7]	14/14
2022- first	15	39.5 [20–50]	302	67.3 [50.0–85.0]	93.6 [85.7–100]	4.8 [3.6–6.6]	15/15
2022- second	21	37.9 [28–60]	590	56.9 [22.7–100]	91.0 [72.7–100]	4.1 [1.8–7.5]	19/21

^1^ Antibody was monitored every six months for 2.5 years. ^2^ The vaccine-induced antibody rate was calculated as the sum of ELISA-positive and ELISA-negative but serum-neutralization-test-positive samples. ^3^ The mean antibody titer was calculated from the actual serum neutralization test titer for ELISA-negative samples and the estimated serum neutralization test titer for ELISA-positive samples. ^4^ Minimum–maximum value of each term. First: April to September; second: October to March next year. ELISA: enzyme-linked immunosorbent assay.

## Data Availability

Data are contained within the article and Appendix A.

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
