# Peer review of "Evaluation of Immune Status of Pigs against Classical Swine Fever for Three Years after the Initiation of Vaccination in Gifu Prefecture, Japan"

_pathogens, 2024, doi:10.3390/pathogens13080616_

Round 1

Reviewer 1 Report

Comments and Suggestions for Authors

In the article entitled: " Evaluation of Immune Status of Pigs against Classical Swine Fever for three years after the Initiation of Vaccination in Gifu Prefecture, Japan " authors provided an interesting article regarding impact of possessing of maternal antibodies and vaccination induced antibodies against Classical Swine Fever (CSF).

Since CSF is economically important disease that may lead to drastic economic losses it is crucial to recognize prevention pathways including effective vaccination. Time of vaccination is especially important. Too early vaccination may interfere with antibodies obtained from vaccination. Too late vaccination expose host to pathogen.  Introduction is sufficient, methods are sound, results are clearly presented, conclusions are supported by results.

Please find below minor comments:

1.      Line 57-58: “Therefore, the timing of vaccination must be re-57 considered. “ I would avoid such conclusion since authors justify it by outbreaks in wild boar. It is normal that pathogen remains or circulate in its natural reservoir (wild boars) while pigs are protected by vaccination. Changing the time of pig vaccination may not change at all the situation of wild boars sine they are not vaccinated. I suggest to write that knowledge of proper time of vaccination will improve the situation of pig herds.

2.      What is the lowest detection limit of ELISA used in this study? Does Table 3 include sum of the results exceeding 1:32 titer or is it all positive results, also with <1:32 titer? Please indicate it to inform if percentage of seropositive animals showed in the 6th column of Table is immunized effectively to combat CSF.

3.      Please clarify why in the Figure 2d, optimal age for vaccination is prolonged to day 18, while maternal antibodies decreased? Is it the error of the estimation model?

4.      Finally It would be very beneficial to try to find pattern of optimal age for vaccination, without the testing of immunity. This pattern should include the time of first vaccination and the generation of sows. Have autors tried to find it?

I’ve got no further comments.

Author Response

Dear editors and reviewers,

Thank you very much for your careful review of our manuscript.

Below, we provide revisions to the points raised by each reviewer.

In addition, the "Institutional Review Board Statement" that was raised by the editor is highlighted in light blue. The minor grammatical errors in English are also highlighted in light blue.

We hope that the revised manuscript is sufficient for publication in Pathogens.

 Response to Reviewer 1:

Thank you for your detailed review. We have addressed the comments as follows. The revised sections are highlighted in yellow in the text.

Comment 1: Line 57-58: “Therefore, the timing of vaccination must be reconsidered. “I would avoid such conclusion since authors justify it by outbreaks in wild boar. It is normal that pathogen remains or circulate in its natural reservoir (wild boars) while pigs are protected by vaccination. Changing the time of pig vaccination may not change at all the situation of wild boars sine they are not vaccinated. I suggest to write that knowledge of proper time of vaccination will improve the situation of pig herds.

[Response]

Thank you for your comment. First, this comment may have been caused by the unclearness of whether the outbreak mentioned in the previous sentence referred to farm pigs or wild boar. We first clarified this point as farm pigs. Based on this, we have clearly stated that “knowledge of the optimal age for vaccination will improve the situation of the pig herd.”

[Revision in the text]

Page 2, lines 53 and 56: “on pig farms” was added.

Page 2, lines 57-59: “Therefore, the immune status of the pig herd may be insufficient, and knowledge of the optimal age for vaccination will improve the situation of the pig herd” was added.

Comment 2: What is the lowest detection limit of ELISA used in this study? Does Table 3 include sum of the results exceeding 1:32 titer or is it all positive results, also with <1:32 titer? Please indicate it to inform if percentage of seropositive animals showed in the 6 column of Table is immunized effectively to combat CSF.

[Response]

The detection limit of ELISA was 1:16, as previously reported (Reference 18). Therefore, ELISA-positive samples had an SNT titer of ³1:16. On the other hand, ELISA-negative samples were tested for SNT. That is, there are three categories of samples: (1) ELISA-positive, (2) ELISA-negative but SNT-positive (≧1:1), (3) both ELISA- and SNT-negative. The vaccine-induced antibody rate was calculated by summing (1) and (2).

[Revision in the text]

Based on your comments, we have added the following statements to Materials and Methods sections 2.2, 2.4, and Table 3 (footnote).

Page 3, line 124: but the lower limit of detection for this ELISA is 1:16 [18].

Page 3, line 149: The vaccine-induced antibody rate was defined as the rate of the total number of either ELISA-positive or ELISA-negative but SNT-positive samples per number of fattening pigs tested on the farm.

 Page 7, Table 3: 2 The vaccine-induced antibody rate was calculated as the sum of ELISA-positive and ELISA-negative but serum neutralization test-positive samples.

Comment 3: Please clarify why in the Figure 2d, optimal age for vaccination is prolonged to day 18, while maternal antibodies decreased? Is it the error of the estimation model?

[Response]

The optimal ages for vaccination shown in Figures 2a, b, c, d, e, and f were 34, 37, 18, 9, and 0 days, respectively. The mean SNT titers of each sow were 9.1, 9.2, 7.9, 7.1, 6.6, and 5.4, which did not prolong but matched the optimal age for vaccination.

[Revision in the text]

No modification in the text.

Comment 4: Finally, it would be very beneficial to try to find pattern of optimal age for vaccination, without the testing of immunity. This pattern should include the time of first vaccination and the generation of sows. Have authors tried to find it?

[Response]

This study identified the optimal age for vaccination based on progress from the start of vaccination. As commented, it would be beneficial to determine the optimal age for vaccination without immunological testing since it is difficult to conduct immunological testing on all farms and to determine the optimal age for each farm. SOW-G1 is easy to identify because of its uniform and low SNT titer variation, whereas SOW-G2 is difficult to find typical pattern because of its high SNT titer variation. Therefore, it is desirable to confirm the optimal age for vaccination by immunological testing for SOW-G2.

[Revision in the text]

Based on your comments, we have added the following statements.

Page 11, lines 390-394: Once these data could be accumulated, the determination of the optimal age for vaccination may be available for SOW-G1 without immunity testing because of the uniform SNT titer without variation; however, for SOW-G2, the optimal age for vaccination should be confirmed by antibody testing because of the variation in SNT titers.

With this revision, the following sentence was added to the next paragraph.

Page 11, lines 395-396: Other concern includes low protection by maternal antibody against field CSFV with diverse antigenicity and moderate pathogenicity.

Page 11, lines 399: Changed "higher/lower" to "higher".

Reviewer 2 Report

Comments and Suggestions for Authors

In the present manuscript, Keisuke KUWATA et al. reported the analysis of antibody statuses in sows and fattening pig herd for 3 years, and the potential optimal vaccination age of piglets was suggested. But, it is well known that immunization with CSFV live attenuated vaccine is interfered by maternal antibody levels. The optimal vaccination age should be determined for different farms separately, by assuming the antibody level in the sows, and more important, in the piglets. In addition, the study just test and analyze the antibody levels, the proposed optimal vaccination age has not been put into practice. Important data is lacking to support the concluded vaccination age.

Author Response

Dear editors and reviewers,

Thank you very much for your careful review of our manuscript.

Below, we provide revisions to the points raised by each reviewer.

In addition, the "Institutional Review Board Statement" that was raised by the editor is highlighted in light blue. The minor grammatical errors in English are also highlighted in light blue.

We hope that the revised manuscript is sufficient for publication in Pathogens.

Response to Reviewer 2:

Thank you for your detailed review. We have addressed the comments as follows. The revised sections are highlighted in green in the text.

Comment: In the present manuscript, Keisuke KUWATA et al. reported the analysis of antibody statuses in sows and fattening pig herd for 3 years, and the potential optimal vaccination age of piglets was suggested. But, it is well known that immunization with CSFV live attenuated vaccine is interfered by maternal antibody levels. The optimal vaccination age should be determined for different farms separately, by assuming the antibody level in the sows, and more important, in the piglets. In addition, the study just test and analyze the antibody levels, the proposed optimal vaccination age has not been put into practice. Important data is lacking to support the concluded vaccination age.

[Response]

Thank you for your comment. Your two points, “The optimal vaccination age should be determined for different farms separately” and “the proposed optimal vaccination age has not been put into practice.” are important.

Regarding your first comment, the optimal age for vaccination should ideally be estimated and determined separately for each sow. Even if estimating the optimal timing of each sow is not feasible, as Reviewer 2 mentioned, the optimal age for vaccination should be determined for each farm. In 2019, vaccination was restarted in Japan on a prefectural basis. Therefore, in this study, changes in the optimal age for vaccination were observed across the entire Gifu Prefecture. It is expected that this method will be applied at the farm- and sow-levels to determine the optimal age for vaccination of piglets.

In terms of your second comment, pigs at three farms (No. 7, 14, and 31) were vaccinated slightly earlier than the estimated optimal age; vaccine-induced antibody rates for these farms were >80%, as presented in Supplement Table S3. Similarly, farm No. 28 was vaccinated at an optimal age and a vaccine-induced antibody rate was >80%. Conversely, no farm had pigs that were vaccinated before the estimated optimal age and vaccine-induced antibody rates was <80%. Thus, we can indicate a few evidence to support our concept. However, the number of evaluated farms was small, and the proposed assertions were limited. Therefore, further analyses should be conducted to validate our proposed optimal vaccination age.

To clarify the above explanation, the optimal vaccination age at each farm and the gap between the optimal and actual vaccination ages are presented in Table S3.

Although the optimal vaccination age is estimated to be before 30 days of age, the factors that prevent vaccination at that time include the following:

  1. The optimal timings for PRRS and PCV-2 vaccinations overlap; therefore, it is necessary to delay the timing of CSF vaccination.
  2. Vaccination at a young age was avoided because of concerns that on-site accidents could occur.

These realities were also included in the discussion to explain the current situation in Japan.

[Revision in the text]

In response to your first comments, we have added the following statement. In addition, we added columns for Optimal age and Gap in vaccination age to Table S3.

Page 4, lines 160: throughout Gifu Prefecture

Page 4, lines177-178: Using this method, the optimal age for vaccination on 45 fattening pig farms was estimated individually based on the SNT data of sows six months prior.

Page 7, lines 266-267: in Gifu Prefecture

Page 7, lines 275-279: The optimal age for vaccination was estimated for 45 farms with sow data. It was evident that most farms were vaccinated later than the optimal age (Table S3). Nevertheless, three farms were vaccinated earlier than the optimal age, and one farm was vaccinated with a gap of 1 day from the optimal age. The vaccine-induced antibody rates at these four farms were >80% (Table S3).

Page 7, lines 280: throughout Gifu Prefecture

Page 11, lines 385-390: Ideally, the optimal age for vaccination should be estimated on a sow-by-sow or farm-by-farm basis. As a first trial, this study was conducted to estimate the optimal age for vaccination throughout Gifu Prefecture area because vaccination in Japan in 2019 was restarted on a prefecture-by-prefecture basis. This approach is also expected to be applied to sow-based and farm-based vaccination programs at the optimal age for vaccination.

Page 11, lines 415-416: Table S3: Vaccine-induced antibody rates and optimal vaccination age of fattening pigs per farm between 2020 and 2022.

In response to your second comment, we have added the following statement.

Page 11, lines 370-382: The method for calculating the optimal age for vaccination based on a previous report [3], which contributed to the eradication of CSF in Japan, was applied to the current antibody situation. Between 2020 and 2022, many farms were vaccinated late, considering the first generation, and the vaccine-induced antibody rate was ³80% and close to 100%. Most importantly, three farms were vaccinated at an earlier than the estimated optimal age, and one farm was vaccinated at the same time as the optimal age, all achieving vaccine-induced antibody rates of ³80% (Table S3). Thus, we can indicate a few evidence to support our concept, estimation of the optimal age for CSF vaccination. However, the number of these practices is small and insufficient to support our assertion. Therefore, it is necessary to increase the number of practices in future. Challenges with CSF vaccination prior to 30 days of age include overstress when vaccinating young pigs and coordination with vaccination programs for other viral diseases. These factors need to be resolved when accumulating data on the evaluation of optimal vaccination timing.

Reviewer 3 Report

Comments and Suggestions for Authors

Dear editor, 

The manuscript entitled "Evaluation of Immune Status of Pigs against Classical Swine Fever for three years after the Initiation of Vaccination in Gifu Prefecture, Japan" by Dr. Kuwata et al., represents data of long study done by researchers over the spam of three years in Japan, where continuous monitoring of Antibodies against CSF were performed in Pig farms to assess the herd immunity status in farms, and the Ab titer in animals through out the vaccination program performed in Japan.

It is known that the immunity (Ab) against CSF is transmitted from mothers to offsprings for the first 30 days, also, the high level of Ab in the blood of previously vaccinated animals can lower the effectiveness of secondary vaccination, or first boost in case of newborn pigs. 

Therefore, it's critical for veterinary authorities, before starting a vaccination program, to monitor the level of Ab in the herd and understand the immunity status so that their programs of vaccination can be effective.

This study, answers these questions and helps the authorities in improving their programs and modifying them for the future. 

Author Response

Dear editors and reviewers,

Thank you very much for your careful review of our manuscript.

Below, we provide revisions to the points raised by each reviewer.

In addition, the "Institutional Review Board Statement" that was raised by the editor is highlighted in light blue. The minor grammatical errors in English are also highlighted in light blue.

We hope that the revised manuscript is sufficient for publication in Pathogens.

Response to Reviewer 3:

Thank you for your detailed review.

Comment: The manuscript entitled "Evaluation of Immune Status of Pigs against Classical Swine Fever for three years after the Initiation of Vaccination in Gifu Prefecture, Japan" by Dr. Kuwata et al., represents data of long study done by researchers over the spam of three years in Japan, where continuous monitoring of Antibodies against CSF were performed in Pig farms to assess the herd immunity status in farms, and the Ab titer in animals through out the vaccination program performed in Japan. It is known that the immunity (Ab) against CSF is transmitted from mothers to offsprings for the first 30 days, also, the high level of Ab in the blood of previously vaccinated animals can lower the effectiveness of secondary vaccination, or first boost in case of newborn pigs. Therefore, it's critical for veterinary authorities, before starting a vaccination program, to monitor the level of Ab in the herd and understand the immunity status so that their programs of vaccination can be effective. This study, answers these questions and helps the authorities in improving their programs and modifying them for the future.

[Response]

Thank you for your comment. Immediately after the start of vaccination in 2018, Japan experienced the emergence of the first-generation sows. This paper provides a reference for taking action in the face of such situation in the future.

[Revision in the text]

There are no additions in the text.

Reviewer 4 Report

Comments and Suggestions for Authors

Keisuke Kuwata et al evaluated of immune status of pigs against CSF for three years after the initiation of vaccination in Gifu Prefecture, Japan. This is a valuable about CSF vaccination.

Major comments:

1.     There is low correlation between ELISA (Nippon Gene Corp., Tokyo, Japan) and SNT in the present study. So I recommend the author can delete the ELISA data from this manuscript, or move to discussion section.

2.     In Gifu Prefecture, vaccination of all pigs started simultaneously on October 25, 2019. Please explain the subsequently sow vaccination program, for example mass vaccination, pre-mating or pre-farrow.

3.     Please explain the vaccination program of SOW-G2

Author Response

Dear editors and reviewers,

Thank you very much for your careful review of our manuscript.

Below, we provide revisions to the points raised by each reviewer.

In addition, the "Institutional Review Board Statement" that was raised by the editor is highlighted in light blue. The minor grammatical errors in English are also highlighted in light blue.

We hope that the revised manuscript is sufficient for publication in Pathogens.

Response to Reviewer 4:

Thank you for your detailed review. We have addressed your comments as follows. The revised parts are indicated with gray highlights in the text.

Comment 1: There is low correlation between ELISA (Nippon Gene Corp., Tokyo, Japan) and SNT in the present study. So I recommend the author can delete the ELISA data from this manuscript, or move to discussion section.

[Response]

Because the antigen immobilized for antibody detection in this ELISA is the E2 protein, the ELISA results correlate with the SNT results (Reference 18). However, the detection limit of this ELISA is 1:16 in the SNT (Reference 18). Therefore, samples with low antibody titers (£1:8) need to be detected by the SNT. As well know, the SNT is complex procedure, so it is efficient to first screening by ELISA, and next performing the SNT on ELISA-negative samples. In Japan, ELISA is performed before SNT based on this concept. From the above, it would be beneficial to the leader to list both the “positive rate by ELISA only” and “positive rate by summing ELISA-positive rate and ELISA-negative but SNT-positive rate”. In response to the reviewer's comment, explanations of the characteristics of ELISA, reasons for its use, and advantages were added to explain the benefits of ELISA.

[Revision in the text]

Based on your comments, we have added the following statement.

Page 3, lines 123-124: This ELISA correlates with SNT because E2 glycoprotein is the main component, but the lower limit of detection for this ELISA is 1:16 [18].

Page 3, lines 126-128: Because SNT is a complex procedure, first screening using convenient ELISA and then performing SNT on ELISA-negative samples are efficient.

Comment 2: In Gifu Prefecture, vaccination of all pigs started simultaneously on October 25, 2019. Please explain the subsequently sow vaccination program, for example mass vaccination, pre-mating or pre-farrow.

[Response]

The vaccination program for sows, based on national guidelines, was administered as an initial vaccination at the same age as fattening pigs (approximately 30 to 60 days old), followed by one vaccination after six months and then every year, totaling four vaccinations. There are no restrictions in the vaccination program for pre-mating or pre-farrowing, but there is a tendency to avoid such vaccinations.

[Revision in the text]

Based on your comments, we have added the following statement.

Page 2, lines 86-90: Sows are vaccinated a total of four times according to the vaccination program, with the same initial vaccination as the fattening pigs, six months after that, and then every other year. Although there are no regulations of the vaccination program for pre-mating and pre-farrowing vaccination, the trend is to avoid these vaccinations.

Comment 3: Please explain the vaccination program of SOW-G2

[Response]

As in Comment 2, SOW-G2 is vaccinated in the same procedure as SOW-G1.

[Revision in the text]

No addition to text because the response to comment 2 was added on page 2, lines 86-90, and line 96 states " SOW-G1 and SOW-G2 even after the implementation of the same vaccination pro-gram."

Round 2

Reviewer 2 Report

Comments and Suggestions for Authors

In this revised version, the author has positively responded (revised the manuscript and gave clarification) to my comments.

Some minor comments are:

1. Line 86-89, the author mentioned the common vaccination procedure of the sows. Did the sows of G1, G1.5 and G2 in this study are vaccinated according to this procedure? This should be described in M&M. The vaccination procedure is important for clarify the high diversity of immune levels in sows and piglets in this study.

2. line 121-124, the ELISA kit is indirect ELISA or competitive/blocking ELISA? The author mentioned that the sensitivity of the ELISA is lower than SNT, why didn't the author choose other commercial kit, such as IDEXX, ID.Vet?

3. line 379-381, to overcome the challenges of CSF vaccination <30 days of age, vaccination could be enhanced (more times) in sows, resulting in high levels of neutralizing antibody both in sows and piglets. In practice, different farms should vaccinated sows more times to reach a high, stable and similar neutralizing antibody levels, so maternal antibody levels in the piglets are also similar and the optimal vaccination age could be easily defined as a short period of time (ie. 35±2d).

4. Table 3 and Figure 2, Besides the positive rate, antibody titer is also important to define the optimal vaccination age.

Author Response

Response to Reviewer 2

We appreciate your detailed review. We have responded to your comments as follows. Revisions have been indicated in red in the text and yellow highlight in Table S2 and S3.

Comment 1: Line 86-89, the author mentioned the common vaccination procedure of the sows. Did the sows of G1, G1.5 and G2 in this study are vaccinated according to this procedure? This should be described in M&M. The vaccination procedure is important for clarify the high diversity of immune levels in sows and piglets in this study.

[Response]

Thank you for your comment. In this study, SOW-G1, G1.5, and G2 were all vaccinated following procedure. Therefore, we have added a new section "2.2 Pig herd vaccination" to the Materials and Methods section to clarify the vaccination protocol.

[Revision in the text]

Page 3, line 108:

Removed mention of vaccine in 2.1 serum samples. ”Pig vaccination on farms and subsequent blood sampling were”→” Blood sampling on farm was”

Page 3, lines 117 and 122:

“2.2. Pig herd vaccination

Pig vaccination using GPE vaccine on farms was performed by veterinarians who were technically trained, as was blood sampling. Sows of all generations were vaccinated for the first time at approximately 30-60 days of age, for the second time six months later, and for the third and fourth time every year thereafter. All fattening pigs were vaccinated at approximately 30-60 days of age with only one vaccine dose.” was added. We reorganized the Materials and Methods numbers accordingly.

Comment 2: line 121-124, the ELISA kit is indirect ELISA or competitive/blocking ELISA? The author mentioned that the sensitivity of the ELISA is lower than SNT, why didn't the author choose other commercial kit, such as IDEXX, ID.Vet?

[Response]

Thank you for your comment. First, the current antibody survey was conducted in each prefecture in accordance with the basic policy of the Ministry of Agriculture, Forestry and Fisheries of Japan. On the other hand, the only ELISA kit that has received national approval in our country is the one presented in this paper. Therefore, we could not conduct evaluations using ELISA kits not approved in our country, such as those from IDEXX.

It should be noted, although this is information before the official announcement, that IDEXX is in the process of obtaining national approval in Japan, and it is expected that evaluations using IDEXX ELISA kit will become possible in the future. However, in the results of the preliminary joint research we conducted with IDEXX, many serum samples with low antibody titers (such as 1:1~1:4) in the neutralization test also tested negative with IDEXX ELISA. Therefore, we believe there is no need to change the argument that ELISA, even when using IDEXX ELISA kit, is less sensitive than the neutralization test.

Although the above will not be mentioned in the paper, we have added the following information to the text: “the ELISA in this study is an indirect ELISA method.

[Revision in the text]

Page 3, line 129:

This ELISA is an indirect ELISA and correlates with SNT because E2 glycoprotein is the main component, but the lower limit of detection for this ELISA is 1:16 [18].

Comment 3: line 379-381, to overcome the challenges of CSF vaccination <30 days of age, vaccination could be enhanced (more times) in sows, resulting in high levels of neutralizing antibody both in sows and piglets. In practice, different farms should vaccinated sows more times to reach a high, stable and similar neutralizing antibody levels, so maternal antibody levels in the piglets are also similar and the optimal vaccination age could be easily defined as a short period of time (i.e. 35±2d).

[Response]

Thank you for your comment. We agree with the basic concept of boosting antibody titers in sows with additional vaccination to align the optimal age of immunity in piglets to around 35 days. On the other hand, for sows vaccinated with the GPE vaccine used in our country, we have already obtained results on the increase in antibody titers from multiple vaccinations as below (Personal communication).

Using 47 sows with low antibody titers, when comparing the neutralizing antibody titers 3 months after the first vaccination and 1 month after the second vaccination, the peak of the antibody titer distribution changed from 1:4 after the first vaccination to 1:32 after the second vaccination. 60% of the pigs responded to the second vaccination with an increase in antibody titers, but the effect was limited and did not reach the high antibody titers of SOW-G1. Thus, while some level of boosting is recognized, it is difficult to reach the levels of 1:256~1:512 of SOW-G1. Therefore, we have added the following methods to adjust the optimal vaccination age to 30-40 days.

  1. Although the effect is limited, a booster is expected with additional vaccination of GPE vaccine (attenuated live vaccine).
  2. Currently, inactivated vaccines are not approved and cannot be used in Japan, but additional vaccination with inactivated vaccines may become an effective method in the future.

[Revision in the text]

Page 11, lines 396-402:

“On the other hand, if it is possible to raise the antibody titer of sows as a herd, the optimal age for vaccination would be returned to 30-60 days of age. As a feasible countermeasure, repeated vaccination of sows should be implemented even though the booster effects of GPE vaccine (attenuated live vaccine) with repeated injections are limited. Alternatively, additional immunological stimulation with an inactivated vaccine may be feasible in future, although no approved inactivated vaccines are currently available in Japan.” was added.

Comment 4: Table 3 and Figure 2, Besides the positive rate, antibody titer is also important to define the optimal vaccination age.

[Response]

Thank you for your comment. To add data on antibody titers to Table 3, it is necessary to measure the neutralizing antibody titers of all 1982 fattening pig samples. Although theoretically possible, the feasibility is low because of limited human resources. To address this issue, we established a method to estimate neutralizing antibody titers from the S/P ratio of ELISA [reference 19], and we calculated the average of the estimated neutralizing antibody titers and added them to Tables S2, S3, and 3.

On the other hand, Figure 2 was calculated based on the neutralizing antibody titers measured in sows; thus, the antibody positivity rate in fattening pigs was not used in this calculation. To clarify this point, some sentences were added in the sections of “Material and Methods” and “Discussion”.

[Revision in the text]

Based on your comments, we added the following statements regarding antibody titers in fattening pigs.

Page 4, lines 159-163:

“The antibody titer of each sample was determined using SNT for ELISA-negative samples. In addition, the SNT titers were estimated for ELISA-positive samples based on the S/P value of the ELISA according to a previous report [19]. These individual data were averaged for all farms for each term.” was added.

Page 6, lines 247-248:

“The mean antibody titers of fattening pigs at each term ranged from approximately 1:16 to 1:128 (Tables 3, S2 and S3)” was added.

Page 7, Table 3:

Added column for mean antibody titer and “3 The mean antibody titer was calculated from the actual serum neutralization test titer for ELISA-negative samples and the estimated serum neutralization test titer for ELISA-positive samples.” was added to the footnote.

Table S2: Estimated antibody titers and footnotes were added.

Table S3:

Average antibody titers per farm using actual and estimated antibody titers and footnotes were added.

In addition, we added the following to clarify that the purpose of calculating antibody-positive rates in fattening pigs differs from the estimation of optimal age for vaccination of piglets using sow antibody titers.

Page 4, line 153:

 “To confirm the effectiveness of the CSF vaccine over the three-year period,” was added.

Page 10, line 346:

 “to confirm vaccine efficacy” was added.

Page 10, lines 366-367:

“Estimating the optimal age for piglets for vaccination based on the distribution of SNT titers in sows,” was added.

Page 13, line 505-507:

Reference 19 was added, and the reference numbers rearranged accordingly.